# Content-Aware Image Resizing Technology Based on Composition Detection and Composition Rules

Bo Wang *, Hongyang Si, Huiting Fu, Ruao Gao, Minjuan Zhan, Huili Jiang and Aili Wang *

School of Measurement-Control Technology and Communications Engineering, Harbin University of Science and Technology, Harbin 150080, China; 1905030316@stu.hrbust.edu.cn (H.S.); 1905030306@stu.hrbust.edu.cn (H.F.); 1905030307@stu.hrbust.edu.cn (R.G.); 1905030326@stu.hrbust.edu.cn (M.Z.); 1905030310@stu.hrbust.edu.cn (H.J.)

* Correspondence: wangboliming@hrbust.edu.cn (B.W.); aili925@hrbust.edu.cn (A.W.);
  Tel.: +86-136-5458-9566 (B.W.)

**Abstract:** A novel content-aware image resizing mechanism based on composition detection and composition rules is proposed to address the lack of esthetic perception in current content-aware resizing algorithms. A composition detection module is introduced for the detection of the composition of the input image types in the proposed algorithm. According to the classification results, the corresponding composition rules in computational esthetics are selected. Finally, the algorithm performs the operations of seam carving using the corresponding esthetic rules. The resized image not only protects the important content of the image, but also meets the composition rules to optimize the overall visual effect of the image. The simulation results show that the proposed algorithm achieves a better visual effect. Compared with the existing algorithms, the proposed algorithm not only effectively protects important image content, but also protects important structures and improves the overall beauty of the image.

**Keywords:** image resizing; content-aware; composition detection; composition rules

## 1. Introduction

With the improvement in the portability of digital image and video capture devices and the rapid development of communication network technology, higher requirements are put forward for displaying and playing digital images and videos on diverse mobile terminals. The size of the displayed image should be adjusted according to its resolution and aspect ratio. In addition, adjusting the image size to meet the user's growing communication speed makes automatic image adjustment an important research field [1]. Traditional image scaling techniques, such as fixed-window cropping and equal-scale scaling, focus on geometric constraints, but do not care about the image content and the overall visual importance. When the image size is adjusted non-proportionally, distortion is inevitable [2,3]. At present, a variety of end users require image resizing techniques to preserve both the important content of the image and the overall visual effect of the image. Content-aware image resizing has become an important field of image research from the image papers published in important journals at home and abroad.

Avidan first proposed seam carving (SC) technology, in what is also the most representative work in the field of content-aware resizing. This algorithm is also known as a backward carving algorithm [4]. The seam-carving algorithm proposed by Avidan in 2007 has the problem of image distortion due to the limitations of energy function definition. In addition, when removing or inserting seams, the algorithm needs to use dynamic programming to find the minimum energy seam, which results in a slow resizing speed. In response to its shortcomings, many scholars have proposed various improved algorithms. Many other forms of improved algorithms based on seam carving have appeared in recent years. For example, image resizing is performed by fusing saliency features such as depth of field information [5]. The addition of image saliency information can avoid the deletion

of important image information. A wall-seam model was proposed [6], which combines saliency features. This algorithm can avoid the deletion of salient information, but distortion occurs in the background area. In [7], the saliency map of the fusion depth of field information was proposed; this improves the edge integrity of the salient region. However, because the depth of field information is not sensitive to the edge, it is easy to destroy the non-main regional structure information.

Resizing based on image warping is another representative algorithm with good processing effect in the field of content-aware resizing technology. The algorithm can be understood as a global optimization problem with constraints. It employs a variety of constraints to the process of image warping, tries to keep the main area of the image from deforming or trying to do scaling, so that the deformation, such as stretching, occurs in a background area with low importance [8]. Although image resizing technology based on image warping can realize the non-proportional scaling of the image without destroying the main content of the image, the algorithm introduces the global optimization problem. Because the calculation amount of the optimization problem is generally large, it needs to be transformed into the least square solution problem, which makes the algorithm not suitable for real-time resizing processing [9]. Therefore, the application scope of content-aware image resizing algorithm based on image deformation is limited, and it is difficult to achieve wide application.

Through the analysis of content-aware image resizing technology based on seam carving and image warping, we can see that most of the current content-aware image scaling algorithms have their own advantages and disadvantages. Therefore, for a single resizing algorithm, it is difficult to obtain satisfactory visual effects for all types of images. Many scholars have proposed combining multiple operators to achieve content-aware image resizing [10–12]. The algorithms proposed above are three main algorithms in the field of content-aware resizing technology. The adjustment strategies adopted by these three algorithms for different regions of image importance are different. The main purpose is to avoid local distortion of the image or destruction of the overall visual effect. Although these algorithms can better achieve non-proportional scaling of images, there are some technical defects. When the image resizing is large, it is easy to cause distortion in the main area of the image or destroy the overall visual effect of the image; when the image content is more complex, it is particularly easy to cause distortion. Many algorithms need to go through a large number of iterative operations in the process of resizing, which has a large computational complexity, takes a long time, and is difficult to use for real-time processing. Therefore, in view of the technical defects of the above three types of algorithm, many scholars have introduced new algorithm ideas based on these algorithms and have proposed some other types of algorithm [13,14]. Therefore, there are different image resizing methods, depending on the image content, that can achieve change in image size while preserving the saliency region [15]. For the problem of operation complexity, some scholars have proposed a fast algorithm suitable for content-aware image resizing [16].

The existing content-aware resizing technology often ignores the overall visual effect while retaining important areas. In the field of computer vision, high esthetics has always been the goal of developers. Although there are some algorithms that do consider esthetic effects, these algorithms are aimed at traditional image cropping. It is difficult to overcome the shortcomings of traditional image cropping algorithms that easily lose important image information [17]. In order to improve the overall beauty and visual effect of an image after resizing, while retaining important areas of the image [18], this paper proposes a content-aware image resizing technology that integrates computable esthetics, addressing the problem of the existing content-aware resizing technology's failure to consider the influence of image esthetics on resizing results. In order to select the corresponding composition optimization module, the corresponding composition detection module is required to detect the composition type of the input image. Only when compositions similar to the input image are detected can the corresponding optimization methods be selected for optimization. This paper uses a composition detection network based on a convolutional

neural network (CNN) proposed in literature [19] to detect the composition type of the input image. Next, the paper further resizes the image detected by the composition detection network. According to the image classification results, corresponding image composition rules are selected to guide the positioning of the significant area in the resizing operation. The important content of the image can be preserved alongside the image size adjustment. The resized image content meets the composition rules as much as possible and improves the visual beauty of the image. Experimental results show that, compared with other resizing algorithms, the proposed algorithm not only preserves important information from the image, but also has more esthetic resizing results.

## 2. Algorithm Description

Since there is no uniform esthetic rule that can be applied to all types of image, it is necessary to classify images. To introduce computable esthetics into content-aware resizing technology, different composition rules should be applied for different composition types to guide subsequent resizing operations. Composition classification is an important research focus in computable esthetics. If you can know the type of composition before processing the image, you can select a specific method for subsequent processing of the image, achieving better results than the general composition rule method. In this paper, the classification network based on CNN proposed in [19] is adopted. The types of composition are divided into the following categories: the rule of thirds, central composition, horizontal composition, symmetric composition, triangular composition, curve composition, vertical composition, right angle composition and pattern composition. The image content studied in this paper is mainly aimed at common landscape photography. Among these nine composition methods, the rule of thirds, central composition, symmetrical composition and horizontal composition rules are commonly used composition methods in landscape photography. For other composition rules, there is no general standard definition, and the composition of visual elements in the image is diversified. Therefore, this paper mainly studies the four composition rules commonly used in landscape photography.

### 2.1. Importance Map Generation Method

Content-aware image resizing results have a high dependence on the definition of image content and importance. It is ideal to select an image importance recognition method that conforms to the characteristics of the human eye. The graph-based visual saliency (GBVS) model proposed in [20] obtains an important region that conforms to 98% of the human eye characteristics, and the model is simple and reasonable. Therefore, the GBVS algorithm is adopted in this paper as the importance map of horizontal composition and symmetrical composition type images to improve accuracy of image content recognition, as shown in Figure 1.

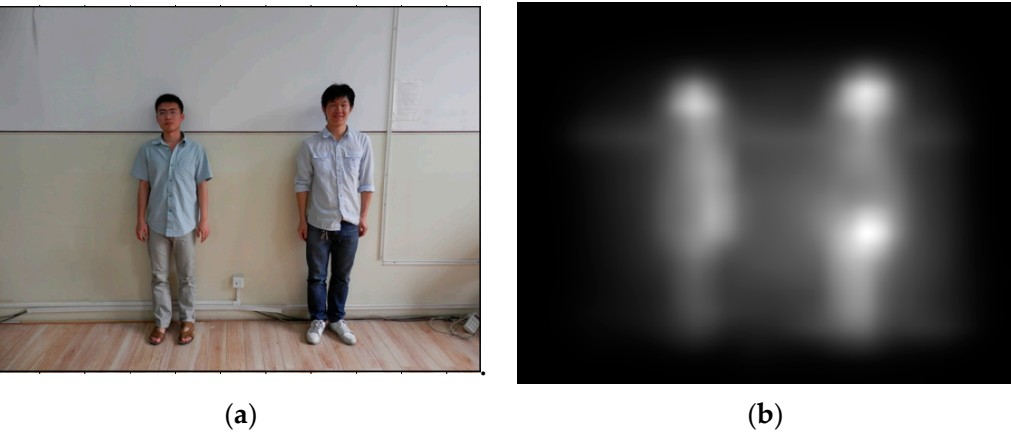

(**a**)　　　　　　　　　　　　　　　　　　(**b**)

**Figure 1.** The generation diagram of GBVS importance map. (**a**) Original image; (**b**) importance map.

However, it is difficult to accurately extract the position of the foreground object for the images of the tripartite composition and the central composition. Therefore, this paper introduces a co-segmentation algorithm when processing the image importance map of the tripartite composition and the central composition [21], which can obtain the foreground image position more accurately. The definition of importance is shown in Formula (1):

$$E_T = E_{\text{grads}} + \alpha E_{seg} \tag{1}$$

A large number of experiments have proved that the value of $\alpha$ is related to the extraction accuracy in the importance map. When set $\alpha = 3$ in this paper, the protection effect of important objects is better, as shown in Figure 2. The importance map obtained by this method can accurately identify the main area of the image.

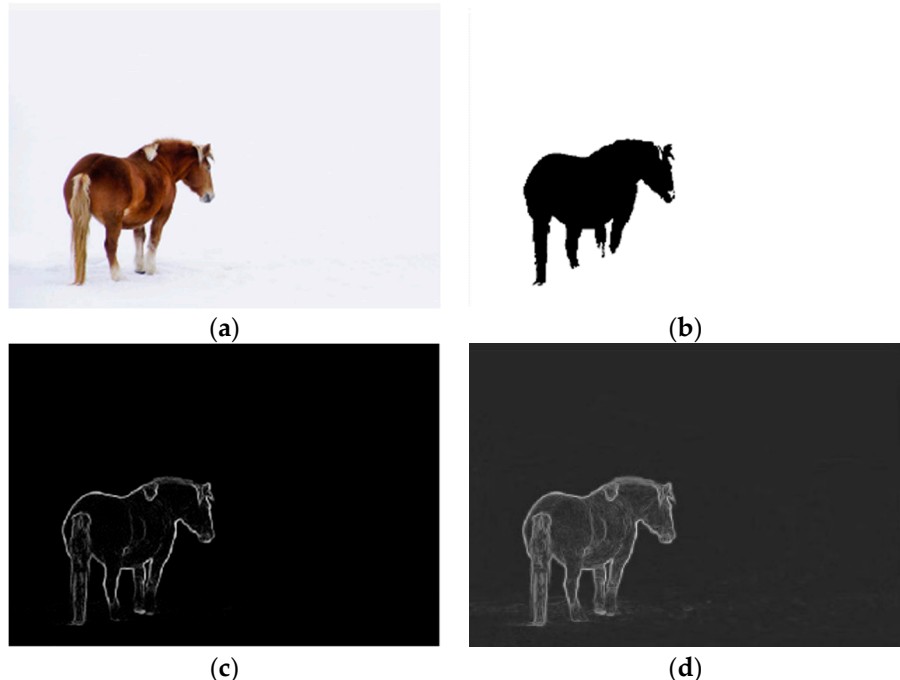

**Figure 2.** The generated schematic diagram of co-segmentation method importance map. (**a**) Original image; (**b**) co-segmentation image; (**c**) gradient map; (**d**) importance map.

## 2.2. Image Resizing Using the Rule of Thirds

For landscape images, the closer the weight ratio of each region in the image is to the golden ratio, the better the visual balance of the image. In photography, when people pay the most attention to the focus of an image, if these concerns are located at the intersection of the three-point line of the image, the image has better visual beauty. In order to attract the attention of the observer, the photographer will adjust the composition of the entire image so that the center of the foreground object is as close as possible to these concerns. Therefore, for a trichotomic composition-type image, S is defined as representing the Euclidean geometric distance between the center point of the foreground object of the original image and the trichotomous point of the target image. S can effectively represent the esthetic value of the type, and the smaller S is, the stronger the image esthetic feeling is.

In order to meet the rules of esthetic composition, the center of the foreground object should be located on the three-point line of the target image in the process of resizing, conforming to the rule of thirds to make the image more esthetic. The four intersections formed by these thirds lines are referred to as "power points". Suppose that the original image size is $W \times H$, the target image size is $W_t \times H_t$, the center point coordinates of the original image are $M(x_m, y_m)$, and the power point coordinates of the target image are $N_i(x_i, y_i)(i = 0, 1, 2, 3)$.

Taking horizontal resizing as an example, the co-segmentation algorithm is used to extract the foreground object of the original image, calculate the coordinates of the center point $M(x_m, y_m)$ of the object, and divide the left and right regions of the original image according to the line $X = x_m$. The Euclidean geometric distance $S = \sqrt{(x_i - x_m)^2 + (y_i - y_m)^2}$ between the center point of the foreground object in the original image and the power point of the target image is calculated, and the minimum distance $S = min\{Euclidean(M, N_i)\}$ is obtained, so as to obtain the power point $N_i(x_i, y_i)$ of the target image closest to the center point of the foreground object in the original image. The number of vertical seams $P_l$ needed to be operated in the left area and $P_r$ needed to be operated in the right area in the original image are calculated, as shown in Formulas (2) and (3). The seams are guided to increase or decrease according to the importance map.

$$P_l = x_m - x_i \tag{2}$$

$$P_r = W_t - W - P_l \tag{3}$$

$P_l$ and $P_r$ can be positive or negative. When the value is positive, the algorithm copies the seams, and deletes the seams when it is negative.

### 2.3. Image Resizing Using Central Composition

Central composition is similar to the rule of thirds. Taking horizontal resizing as an example, it is assumed that the original image size is $W \times H$, the target image size is $W_t \times H_t$, the center point of the original image is $M(x, y)$, and the center point of the target image is $M(x_c, y_c)$. The foreground object of the original image is extracted by the co-segmentation algorithm, and the coordinates of the center point of the object are calculated. The left and right regions of the original image are divided according to the straight line $X = x$. The number of vertical seams $P_l$ needed to be operated in the left area and $P_r$ needed to be operated in the right area in the original image are calculated, as shown in Formulas (4) and (5). The seams are guided to increase or decrease according to the importance map.

$$P_l = x - x_c \tag{4}$$

$$P_r = W_t - W - P_l \tag{5}$$

$P_l$ and $P_r$ can be positive or negative. When the value is positive, the algorithm copies the seams, and deletes the seams when it is negative.

### 2.4. Image Resizing Using Horizontal Composition

For horizontal composition images, the closer the visual weight ratio of each area is to the golden ratio 0.618, the better the visual effect of the image is. For this type of image, the semantic segmentation algorithm is used to obtain the semantic horizontal line of the original image, $y = l$; the upper and lower heights are $L_u$ and $L_d$, respectively. The semantic horizontal line of the target image is $y = l_t$, and the upper and lower heights are $L_{ut}$ and $L_{dt}$, respectively. The original image and the target image are segmented into upper and lower regions, respectively. The number of horizontal seams $H_u$ that need to be operated in the upper region and the number of horizontal seams $H_d$ that need to be operated in the lower region in the original image are calculated as shown in Formulas (6) and (7). The increase and decrease of the seam is guided according to the importance map and the upper and lower height ratio is set to 0.618, which is in line with the golden section ratio, as shown in Formula (8).

$$H_u = L_u - l_{ut} \tag{6}$$

$$H_d = W_t - W - H_u \tag{7}$$

$$0.618 = \frac{L_u + H_u}{L_d + H_d} \tag{8}$$

$H_u$ and $H_d$ can be positive or negative. When the value is positive, the algorithm copies the seams, and deletes the seams when it is negative.

*2.5. Image Resizing Using Symmetric Composition*

Symmetric composition is similar to horizontal composition. Taking the vertical direction as an example, the semantic segmentation algorithm is used to obtain the semantic vertical line $x = k$ of the original image, and the widths of the left and right sides are $O_l$ and $O_r$, respectively. The semantic vertical line $x = k_t$ of the target image is obtained, and the widths of the left and right sides are $O_{lt}$ and $O_{rt}$, respectively. The original image and the target image are divided into left region and right region, respectively. The number of vertical seams $H_l$ that need to be operated in the left region of the original image and the number of vertical seams $H_r$ that need to be operated in the right region are calculated as shown in Formulas (9) and (10). According to the importance map, the increase or decrease of the seams is guided, so that the left and right width ratio is 1, which conforms to the symmetrical ratio 1, as shown in Formula (11).

$$Q_l = O_l - O_{lt} \tag{9}$$

$$Q_d = W_t - W - H_l \tag{10}$$

$$1 = \frac{O_l + H_l}{O_r + H_r} \tag{11}$$

$H_l$ and $H_r$ can be positive or negative. When the value is positive, the algorithm copies the seams, and deletes the seams when it is negative.

## 3. Simulation Experiment and Performance Analysis

In order to test the performance of the proposed algorithm, the proposed algorithm was compared with the cutting algorithm based on esthetics [22] and the SC algorithm [4]. In this paper, the simulation experiment was run on the PC platform with Intel(R) Core (TM) i5-9300H CPU @ 2.40 GHz and 8 GB memory. The composition detection network trained on a KU_PCP dataset [19] was selected as the composition detection module in this paper. The composition detection module was used to classify the images in the data set, and the corresponding esthetic principles were selected to resize the images after classification. To ensure the effectiveness of the algorithm, we implemented our method on the image library [23]. We explain the scheme with the case of image reduction. To make the results more persuasive, the images used in the experiment have significantly attributed lines/edges, foreground objects, faces/people, texture and geometric structures.

The image resizing with the rule of thirds is shown in Figure 3, and the image size is reduced from $1024 \times 683$ to $768 \times 683$. It can be seen from Figure 3b that leaf information on the left side of the image and some fence information on the right side of the image are lost in the image obtained by the esthetics cutting algorithm. In Figure 3d, the house is located on the left three-point line of the image, and the main object located on the three-point line should not be distorted or deformed as far as possible. In Figure 3c, the wall and chimney on the left side of the house are obviously deformed. Compared with Figure 3c, Figure 3d is more esthetic and more consistent with the original image information.

The image resizing with central composition is shown in Figure 4, and the image size is reduced from $1024 \times 673$ to $512 \times 673$. It can be seen from Figure 4b that the image obtained by the esthetic-based cutting algorithm loses part of the information of the car in the background. Compared with Figure 4c, more information of the background car is retained in Figure 4d. In Figure 4c, some wheel information is lost and deformed. Figure 4d retains the relevant information of the wheel as much as possible after resizing, and avoids distortion. From the perspective of esthetic composition, it is more esthetic and more consistent with the original image composition.

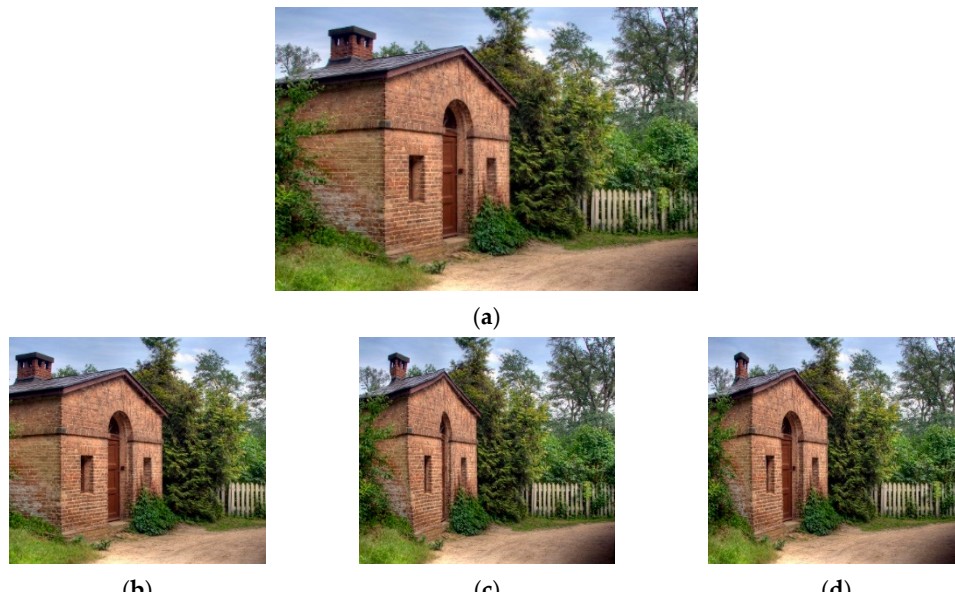

**Figure 3.** Comparison of image reduction using the rule of thirds. (**a**) Original image; (**b**) cutting; (**c**) SC; (**d**) proposed algorithm.

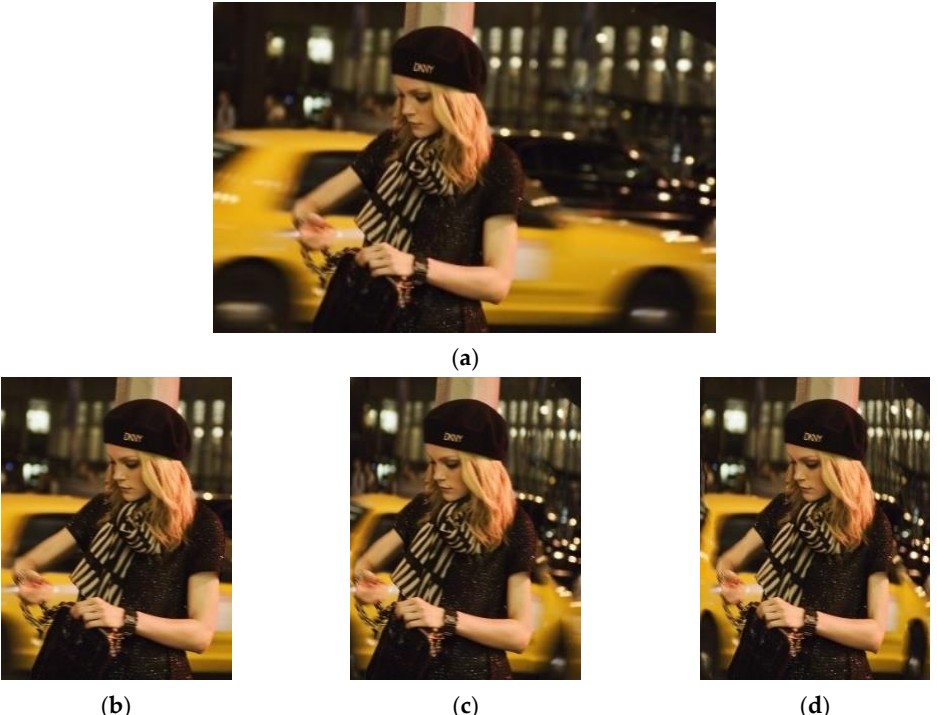

**Figure 4.** Comparison of image reduction using center composition. (**a**) Original image; (**b**) cutting; (**c**) SC; (**d**) proposed algorithm.

The image resizing with horizontal composition is shown in Figure 5, and the image size is reduced from $568 \times 426$ to $568 \times 320$. It can be seen from Figure 5b that the image obtained by the esthetic-based cutting algorithm loses the sky information mapped by the water surface. Compared with Figure 5c, Figure 5d retains the proportion of water surface and sky of the original image. From the perspective of esthetic composition, the visual weight ratio of each region is closer to the golden ratio, and the visual balance effect is better.

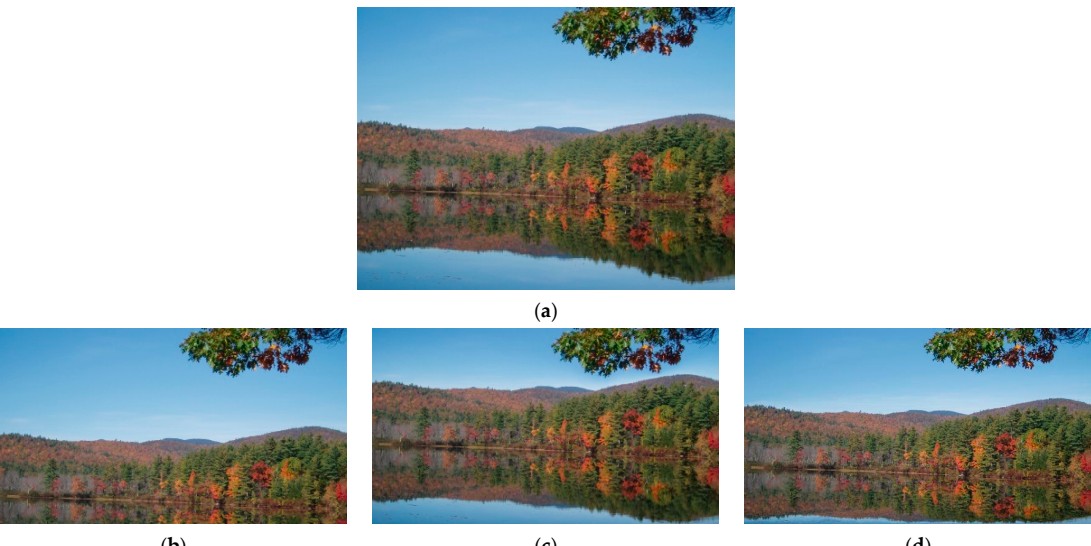

**Figure 5.** Comparison of image reduction using horizontal composition. (**a**) Original image; (**b**) cutting; (**c**) SC; (**d**) proposed algorithm.

The image resizing with symmetric composition is shown in Figure 6, and the image size is reduced from $1024 \times 681$ to $512 \times 681$. It can be seen from Figure 6b that the image obtained by the esthetic-based cutting algorithm loses some information on the right side. Compared with Figure 6c, from the perspective of esthetic composition, Figure 6d is more in line with the symmetrical proportion and avoids image distortion under the condition of maintaining visual balance, as can be seen from the back of the recliner in the picture. In Figure 6c, the image information is distorted, and the four people on the recliner and the banner on the pillar are obviously distorted.

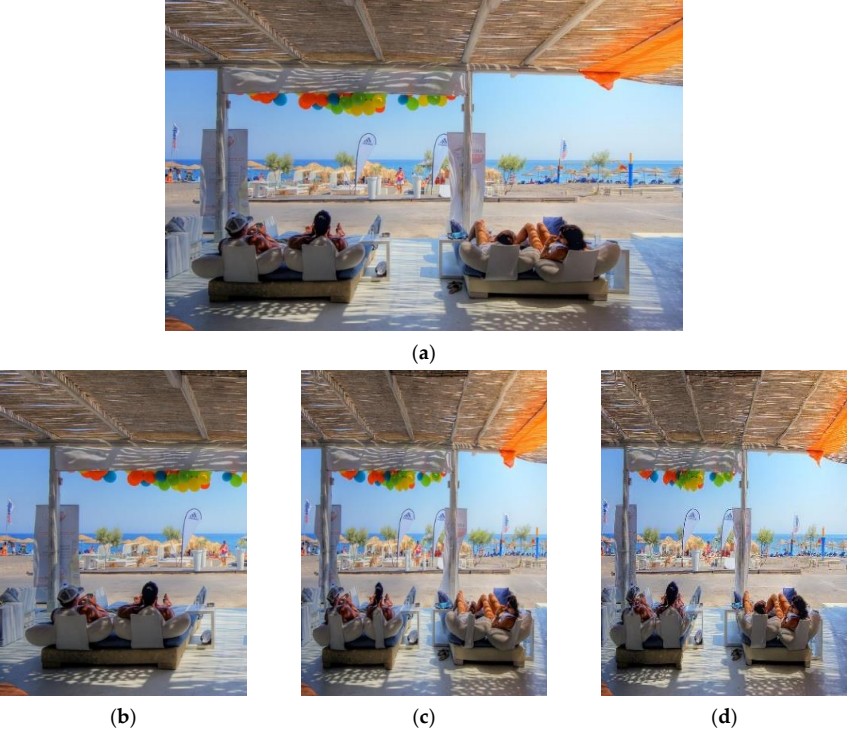

**Figure 6.** Comparison of image reduction using symmetric composition. (**a**) Original image; (**b**) cutting; (**c**) SC; (**d**) proposed algorithm.

The objective performance of the algorithm proposed in this paper was further verified using the evaluation indicators proposed in reference [24]. This evaluation metric calculates the quality of the resized image by combining geometric distortion of and information loss from the image. Specifically, the size of the information loss value represents the proportion of visually significant content lost during the resizing process. The size of the geometric distortion value represents the deformation size of significant objects during the resizing process. The range of evaluation indicators is [0, 1], and the larger the value, the better the image quality. Table 1 shows the image quality index after resizing using different algorithms. Compare the algorithm proposed in this paper with the SC algorithm. The quality index of the proposed algorithm is higher than that of the SC algorithm, because the algorithm in this paper can better retain the visually significant part of the image and reduce information loss, so it can obtain a better quality index.

**Table 1.** Comparison of different image quality indexes.

| Image | Size Change | SC | Proposed Algorithm |
| --- | --- | --- | --- |
| Figure 3 | 25% | 0.767 | 0.788 |
| Figure 4 | 50% | 0.721 | 0.735 |
| Figure 5 | 25% | 0.812 | 0.825 |
| Figure 6 | 50% | 0.694 | 0.722 |

## 4. Conclusions

This paper proposes a content-aware image resizing mechanism based on composition detection and composition rules. The composition detection module is introduced to detect the composition of the input image types in the proposed algorithm. For landscape images, the images are divided into four common composition types by classification method. According to the classification results, the corresponding composition rules in computational esthetics are selected. Finally, the composition rules in computable esthetics are used to guide the resizing operation process. The algorithm can improve the overall visual effect of the image while ensuring that the main content of the image is not distorted, so that the resized image has a high sense of beauty. The experimental results show that, compared with similar algorithms, the algorithm proposed in this paper can achieve ideal results for the scaling of landscape images; it is more esthetic while retaining the important information of the original image.

The algorithm in this paper also has some shortcomings. It mainly studies four composition rules commonly used in landscape photography. Applying the algorithm in this paper to more kinds of image resizing and expanding the applicable scope is the next research direction. It is not enough to provide subjective evaluations when evaluating whether a resized image is more esthetically pleasing while retaining important content. In future research, it is hoped that, based on a given input image, a detailed analysis of the resized image can be conducted from perspectives such as composition, color, light and shadow. At the same time, combined with the image text description, based on esthetic detailed analysis of the image, the content and esthetic attributes of the image can be described naturally, so as to achieve a detailed evaluation of the image. According to the results of detailed evaluation, specific optimizations are made to the composition, color, contrast, and other angles of the image, in order to ensure the content of the image while enhancing its esthetic appeal.

**Author Contributions:** Conceptualization, B.W., H.S. and A.W.; methodology, B.W., H.S. and H.F.; software, H.S. and R.G.; validation, B.W. and M.Z.; writing—original draft preparation, B.W., H.S. and H.J.; writing—review and editing, B.W. and H.S.; supervision, project administration, B.W. and A.W. All authors have read and agreed to the published version of the manuscript.

**Funding:** This research was funded by Heilongjiang Provincial Natural Science Foundation of China, grant number YQ2022F014.

**Data Availability Statement:** No new data were created or analyzed in this study. Data sharing is not applicable to this article.

**Conflicts of Interest:** The authors declare no conflict of interest.

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
