# Peer review of "Content-Aware Image Resizing Technology Based on Composition Detection and Composition Rules"

_electronics, doi:10.3390/electronics12143096_

Round 1

Reviewer 1 Report

The authors propose a content-aware image resizing mechanism based on composition detection and composition rules.

The overall idea seems interesting, but numerous questions related to the scientific soundness and methodology should be answered:

1) The authors propose a CNN-based method ( [16] ), but no real details of the architecture are provided. Why?

2) CNN is often used for classifying; thus, how can they generate images with resizes? It is totally unclear to me. If a generator is involved, an AE is expected, or something similar… Moreover, there is no information about the training (ablation study) and hardware architecture of the involved GPU.

3) No procedure pipeline is shown; can you introduce it?

4) No reference dataset is involved. Why?

5) No comparisons with other similar systems in the state-of-the-art are shown. Why?

The manuscript deals with an interesting problem, but there are some methodological issues that should be fixed.

Minor changes:

The abstract is not clear: the proposal of the authors is not exalted as it should. We suggest underlining the salient features of the work.

Formulas inside the text are incorrectly managed. They should be fixed and restyled.

There are numerous typos, from missing spaces to wrong punctuation management.

We also suggest revising the English to improve the formalization of the sentences, which are currently a bit far from the standard of a scientific paper.

English is quite good; however, some sentences and terms could improve fluency. We suggest having special attention to this aspect of the manuscript.

Reviewer 2 Report

This paper presents a novel approach for content-aware image resizing that leverages composition detection and composition rules. Specifically, the mechanism focuses on landscape images and employs a classification method to categorize them into four common composition types. By incorporating computable aesthetics and utilizing composition rules, the resizing process is guided to improve the overall visual effect of the image. The algorithm ensures that the main content of the image remains undistorted, resulting in resized images with a heightened sense of beauty. Experimental results demonstrate that the proposed algorithm achieves excellent outcomes when resizing landscape images across the four frequently encountered composition types.

Overall, I think this paper is interesting and sounds reasonable. However, this proposed method is not a deep learning-based method, which seems heuristic. Besides, there are some important references about salient region detection missing:

Context-Aware Graph Label Propagation Network for Saliency Detection. TIP 2020.

Deep group-wise fully convolutional network for co-saliency detection with graph propagation. TIP 2019.

line 164 there should be a space between is?. The authors should carefully check the grammar mistakes.

Reviewer 3 Report

Review comments on “Content-aware image resizing technology based on composition detection and composition rules” by Bo Wang etl.

This work presents a content-aware image resizing mechanism based on composition detection and composition rules.

My main general comments are as below:

- An important shortcoming is that the author does not highlight the contribution of their manuscript in comparison to the work that has been performed by previous researchers. This can be added in the introduction and/or conclusion section.

- Conclusions need more elaboration about: outcomes, limitations, and possible/future scenarios.

- The authors should extremely investigation the quantitative and qualitative metrics and compares with the proposed method. The authors should investigate experimentally the quantitative and qualitative metrics of the proposed method.

- The authors should share the proposed method.

Reviewer 4 Report

Review:

The Abstract text has to be paraphrased, and the writing quality needs to be improved (lines 15, 16).
The qualitative and quantitative impact of the study is not mentioned in the paragraph on methodology.
Lines 133 and 134 of the text include formulas that are not height aligned with the text.
The text contains numerous typos, words that are not evenly spaced, and sentences that have extra full stops at the conclusion. Proofreading for spelling is necessary. Usually, the description sentences underneath the image begin with a lowercase letter.
There is a lack of additional explanations of the algorithm's key operations, which affect the composition and scale of the image. The adjustment of function parameters, correlation or influencing parameters, and how the program-software can be tweaked and managed to the desired form and to obtain a better solution and outcome, are not shown in any graphs or tables of comparison.
Additionally, it lacks additional examples and graphical displays of those examples. There isn't enough Open Access reference literature and material in that field to allow for the comparison of studies and the tracking of the scientific method.

Valid research evidence is either not indicated or is not described in a valid manner in the conclusion chapter.
As a result, the paper's thesis and the seven authors' individual contributions, who are listed in the paper's heading, are unclear.
The beginning of this scientific paper is intriguing, but the presentation of the accomplishment is inadequate and ambiguous. So that the research is accurately defined, presented, connected, similar, and ultimately verifiable or replicated in other scientific circles, I advise augmenting with examples, tuning coefficients, progress projection in visual and tabular form.

Finally, the paper's poor writing distracts the reader and lessens the significance of the findings.

The text must be scientifically adjusted to the desired level and paraphrased.

The entire document has to be edited and paraphrased.

Reviewer 5 Report

The percentage of classification should be addressed.

Numerical data as output of different methods should be compared.

The quality of the paper is acceptable.

Reviewer 6 Report

The paper propose a new method for content-aware of natural image. The core idea is well supported with fundamental concept. However, the technical flow in the paper must be improve. There are many places with insufficient clarity and definition of variables. For example, it is not clear why you choose "0.618" in eq(8) and "1" in eq(11). Adding the table with algorithmic steps could help to the readers.

Authors are encourage to cite/discuss the follow relevant works:

a) "Fast scale-adaptive bilateral texture smoothing," IEEE Transactions on Circuits and Systems for Video Technology, vol. 30, no. 7, pp. 2015 - 2026, 2020.

b) "Saliency guided image detail enhancement," Proc. National Conference on Communications (NCC), Bangalore, India, 2019.

Needs to improve significantly.

Reviewer 7 Report

There are several editorial issues. E.g., ‘he proposed algorithm’. The manuscript has a low integrative value in the current debates on the topic, as most cited sources are quite old. Sometimes even the cumulated sources are too old to reflect the current picture. The reference list is too short to cover the topic adequately. The manuscript will benefit from further discussion of key concepts and methodological criteria in order to offer a better articulation between theory and data. ‘in important journals at home and abroad’ – vague. ‘Israel professor’ – avoid such phrases. ‘Shai Avidan et al. has the problem’ – the year must be mentioned for the cited source, remove the first name. Why using certain words and phrase alternatively in initial capital letter? E.g., ‘field of Content-aware image resizing’,  ‘based on Seam Carving have appeared’. More development and depth of the methodology and analysis are needed. ‘Reference [5] proposed’ – unusual. Replace it/they with the proper words to avoid confusion. E.g., ‘It introduces a variety’. ‘have introduced new algorithm ideas based on these algorithms and proposed some other types of algorithms’ – avoid close word repetitions. ‘based on a lot of analysis and research’ – remove this, zero info. ‘CNN’ must be written first in full. ‘Make the resized image content as much as possible to meet the composition rules, improve the visual beauty of the image.’ – poorly constructed. ‘Composition classification is an important research focus in computable aesthetics’ – say directly why. ‘proposed in Reference [16] is’ – ‘proposed in [16] is’. ‘proposed in Reference [17] obtains’ – ‘proposed in [17] obtains’. The figures should be improved and thoroughly explained. You should compare your results with others in terms of concrete data for better research integrative value. The discussions require more structure and there is a need of offering a clear assessment of reviewed literature. Some cited sources are not developed. The conclusions drawn, too short, are not well justified in the data collected and should clarify the main contribution of the paper and the value added to the field.
The relationship between remote sensing and image recognition technologies and remote big data management and visual imagery tools as regards content-aware image resizing mechanisms based on composition detection and composition rules has not been covered, and thus such sources can be cited:
Nica, E., and Vahancik, J. (2023). “Geospatial Big Data Management and Computer Vision Algorithms, Remote Sensing and Image Recognition Technologies, and Event Modeling and Forecasting Tools in the Virtual Economy of the Metaverse,” Linguistic and Philosophical Investigations 22: 9–25. doi: 10.22381/lpi2220231.
Zvarikova, K., Rowland, Z., and Nica, E. (2022). “ Multisensor Fusion and Dynamic Routing Technologies, Virtual Navigation and Simulation Modeling Tools, and Image Processing Computational and Visual Cognitive Algorithms across Web3-powered Metaverse Worlds,” Analysis and Metaphysics 21: 125–141. doi: 10.22381/am2120228.
Woodward, B. (2023). “Remote Big Data Management and Visual Imagery Tools, Multisensor Fusion and Dynamic Routing Technologies, and 3D Space Mapping and Object Recognition Algorithms on Blockchain-based Metaverse Platforms,” Linguistic and Philosophical Investigations 22: 60–76. doi: 10.22381/lpi2220234.

There are several editorial issues. E.g., ‘he proposed algorithm’. ‘Make the resized image content as much as possible to meet the composition rules, improve the visual beauty of the image.’ – poorly constructed. Why using certain words and phrase alternatively in initial capital letter?

Round 2

Reviewer 4 Report

The lack of reference literature, particularly available through Open Accesss, is still a major drawback of this work, which diminishes the originality and traceability of the research.

The methodology is still insufficiently precise and incomplete in terms of achieving better results thanks to the innovative planned improvement of the procedure.

Insufficiently clear images selection and images description. Scarce discussion in content and in terms of connectivity with research.
The conclusion should be clearer and more focused on the settings and improvements obtained during the research and procedure.

Some errors in text and spelling still present. Correct proofreading is necessary.

Reviewer 5 Report

Incorporated the comments with justifications. 

Minor typesetting needs to be done as per the MDPI format.

Reviewer 7 Report

The manuscript is still poorly written, edited, and documented, and the added content makes the situation worse. E.g.,

Avidan first proposed seam carving (SC) technology, which is also the most representative work in the field of content-aware image resizing. This algorithm is also known as backward Carving algorithm [3]. In 2007, the seam carving algorithm proposed by Avidan et al. has the problem of image distortion due to the limitation of energy function definition. Moreover, each time the seam is removed or inserted, the algorithm needs to use dynamic programming to find the minimum energy seam, resulting in slower resizing speed. In view of its shortcomings, many scholars have proposed a variety of improved algorithms. Many other forms of improved algorithms based on seam carving have appeared in recent years. For example, image resizing is performed by fusing saliency features such as depth of field information [4].

The experimental results show that compared with similar algorithms, the proposed algorithm can achieve ideal results for landscape image scaling of four commonly used com-position types, and it is more aesthetic while retaining the important information of the original image.

It is not enough to provide subjective evaluations when evaluating whether a resized im-age is more aesthetically pleasing while retaining important content.

The manuscript is still poorly written, edited, and documented, and the added content makes the situation worse. E.g.,

Avidan first proposed seam carving (SC) technology, which is also the most representative work in the field of content-aware image resizing. This algorithm is also known as backward Carving algorithm [3]. In 2007, the seam carving algorithm proposed by Avidan et al. has the problem of image distortion due to the limitation of energy function definition. Moreover, each time the seam is removed or inserted, the algorithm needs to use dynamic programming to find the minimum energy seam, resulting in slower resizing speed. In view of its shortcomings, many scholars have proposed a variety of improved algorithms. Many other forms of improved algorithms based on seam carving have appeared in recent years. For example, image resizing is performed by fusing saliency features such as depth of field information [4].

The experimental results show that compared with similar algorithms, the proposed algorithm can achieve ideal results for landscape image scaling of four commonly used com-position types, and it is more aesthetic while retaining the important information of the original image.

It is not enough to provide subjective evaluations when evaluating whether a resized im-age is more aesthetically pleasing while retaining important content.

Round 3

Reviewer 7 Report

The manuscript is still extremely poorly written, edited, and documented. Here are some examples (there are dozens):

Avidan first proposed seam carving (SC) technology, which is also the most representative work in the field of content-aware image resizing. This algorithm is also known as backward Carving algorithm [3]. In 2007, the seam carving algorithm proposed by Avidan had the problem of image distortion due to the limitation of the definition of the energy function.

In order to improve the overall aesthetic and visual effect of the resized image while preserving the important areas of the image [16], this paper proposes a content-aware image resizing technology that combines computable aesthetics

To select the corresponding composition optimization module, it is necessary to have the corresponding composition detection module to detect the composition type of the input image. Only when a similar com- position of the input image is detected, the corresponding optimization method can be selected for optimization.

It is not enough to provide subjective evaluations when evaluating whether a resized im-age is more aesthetically pleasing while retaining important content.

The manuscript is still extremely poorly written, edited, and documented. Here are some examples (there are dozens):

Avidan first proposed seam carving (SC) technology, which is also the most representative work in the field of content-aware image resizing. This algorithm is also known as backward Carving algorithm [3]. In 2007, the seam carving algorithm proposed by Avidan had the problem of image distortion due to the limitation of the definition of the energy function.

In order to improve the overall aesthetic and visual effect of the resized image while preserving the important areas of the image [16], this paper proposes a content-aware image resizing technology that combines computable aesthetics

To select the corresponding composition optimization module, it is necessary to have the corresponding composition detection module to detect the composition type of the input image. Only when a similar com- position of the input image is detected, the corresponding optimization method can be selected for optimization.

It is not enough to provide subjective evaluations when evaluating whether a resized im-age is more aesthetically pleasing while retaining important content.

Author Response

Point 1:The manuscript is still extremely poorly written, edited, and documented. Here are some examples (there are dozens):

Avidan first proposed seam carving (SC) technology, which is also the most representative work in the field of content-aware image resizing. This algorithm is also known as backward Carving algorithm [3]. In 2007, the seam carving algorithm proposed by Avidan had the problem of image distortion due to the limitation of the definition of the energy function.

In order to improve the overall aesthetic and visual effect of the resized image while preserving the important areas of the image [16], this paper proposes a content-aware image resizing technology that combines computable aesthetics

To select the corresponding composition optimization module, it is necessary to have the corresponding composition detection module to detect the composition type of the input image. Only when a similar com- position of the input image is detected, the corresponding optimization method can be selected for optimization.

Response 1: 

Avidan first proposed seam carving (SC) technology, which is also the most representative work in the field of content-aware resizing. This algorithm is also known as backward carving algorithm [3]. In 2007, the seam carving algorithm proposed by Avidan has the problem of image distortion due to the limitations of energy function definition.

In order to improve the overall beauty and visual effect of the image after resizing while retaining important areas of the image[16], this paper proposes a content-aware image resizing technology that integrates computable aesthetics.

In order to select the corresponding composition optimization module, the corresponding composition detection module is required to detect the composition type of the input image. Only when compositions similar to the input image are detected, the corresponding optimization methods can be selected for optimization.

I have reviewed the full text and revised it one by one.

Point 2:It is not enough to provide subjective evaluations when evaluating whether a resized im-age is more aesthetically pleasing while retaining important content. 

Response 2:

In the second round of modifications, I have added objective evaluations as required.

Part 3, Page 9, Paragraph 1.

The objective performance of the algorithm proposed in this paper was further veri-fied using the evaluation indicators proposed in reference [22]. This evaluation metric calculates the quality of the scaled image by combining geometric distortion and infor-mation loss of the image. Specifically, the size of the information loss value represents the proportion of visually significant content loss during the resizing process. The size of ge-ometric distortion values represents the deformation size of significant objects during the resizing process. The range of evaluation indicators is [0,1], and the larger the value, the better the image quality. Table 1 shows the image quality index after scaling using different algorithms. Compare the algorithm proposed in this paper with the SC algorithm. The quality index of the proposed algorithm is higher than that of the SC algorithm, because the algorithm in this paper can better retain the visually significant part of the image and reduce information loss, so it can obtain a better quality index.

Table 1. Comparison of different image quality index.

Image

Size change

SC

Proposed algorithm

Figure 3

25%

0.767

0.788

Figure 4

50%

0.721

0.735

Figure 5

25%

0.812

0.825

Figure 6

50%

0.694

0.722